# Optimization of a Laboratory Rainfall Simulator to Be Representative of Natural Rainfall

**María Fernández-Raga** [1,*], **Indira Rodríguez** [1], **Pablo Caldevilla** [2], **Gabriel Búrdalo** [1], **Almudena Ortiz** [2] **and Rebeca Martínez-García** [2]

1 Department of Chemistry and Applied Physics, University of León, 24071 León, Spain
2 Department of Mining Technology, Topography and Structures, University of León, 24071 León, Spain
* Correspondence: maria.raga@unileon.es

**Abstract:** The importance of understanding the effects of rainfall on different materials over time makes it essential to carry out controlled tests to reduce analysis time. Rainfall simulators have been in use for decades and have been implemented as technology and knowledge of the physical behavior of water advanced. There are two main types of rainfall simulators: gravity simulators and pressure simulators. In the former, the drop velocity is normally smaller than the terminal velocity reached by natural droplets; in the latter, the drop size is too small to be representative and has far more speed than the natural speed for those sizes. To solve this problem, a simulator has been developed where the terminal velocity of the raindrops is reached and the drop size can be varied by different nozzles of variable sizes, adapting it to the conditions of a given region. In this study, conditions similar to the rainfall conditions of the city of León have been achieved. This paper presents the design of a rainfall simulator that recreates different rainfall conditions and rainwater composition and its calibration process.

**Keywords:** rainfall simulator; drop size distribution; terminal speed



## 1. Introduction

Rainfall simulators are highly valuable instruments in research because of their versatility. They appeared for the first time in the bibliography in 1966, and since then, they have been used in very different areas. For example, they have been used for measuring the erosion of soil [1–10], measuring the impact of rainfall over monuments [11,12], measuring the loss of productivity of burnt soil [13], assessing harm caused by drops in the leaves of cultivation areas [14], estimating the number of fertilization products used in agriculture [15], measuring the attenuation of radio waves by a storm [16], measuring water infiltration in concrete structures [17], evaluation of resistance of unpaved roads to understand the hydrologic processes affecting urban drainage systems [18–20], etc. Nevertheless, in order to extract the most out of this technique, it is important to ensure it is representative of natural rainfall, focusing especially on the correct representation of kinetic energy as the main characteristic of rainfall. After thorough research has been conducted about their validity as an instrument, it has been proven that currently there are no perfect rainfall simulators to represent natural rainfall because some of them have not reached a real drop size distribution, some produce drops that are faster that the terminal speed of the natural drop for the same size, and others produce drops bigger than the those in natural rainfall.

One possible classification of rainfall simulators depends on the places where they will be used because depending on the purpose of the research, they can be designed as portable rainfall simulators [21] or fixed ones installed in a laboratory [22]. Obviously, the fixed simulators normally characterize rainfall more accurately [7] because in a portable design, their precision is compromised by the lightness and small size required to install it in what can be very inaccessible places (e.g., burnt areas, agricultural lands). In typical fixed

laboratory simulators, there are some characteristics that are sacrificed, such as reaching the terminal speed per size of the drops or the use of distilled water, as is the case with natural rainfall [5]. In the laboratory rainfall simulators, the type of water and the height the drops falls from can be controlled [23] but most of the time these characteristics are sacrificed because of the difficulties associated with adapting the size of the laboratories where they must be installed or the slow and expensive production of distilled water for their functioning.

Another classification considers the working system of those simulators, which may be gravity or pressure-driven systems. In the gravity-driven system, the drip method is used to produce rainfall. As a result, the drops are too large [18] because normally the drops are formed naturally after passing through a hole of different size on the bottom part [20,23]. This means that no pump is used; therefore, no additional energy supply is needed and the water is accumulated on the top of the rainfall simulator to reach the required energy to produce the rainfall. These water droplets are very heavy, and this means that the total quantity of rainfall is limited. This limitation makes working with these rainfall simulators in the field more difficult because the structure required to support a stand with such height and the water deposit on top is too heavy. Another drawback is that the total height of such rainfall simulators is shorter and if there is less than 10 m from the drop production area, it is not possible for the drops to reach the terminal speed [2,24–27]. Furthermore, the drops produced by this rainfall simulator usually show homogenous sizes alongside the sample area.

Regarding the pressure rainfall simulators, they do not have this limitation in the weight of the structure because the water can be pumped from a deposit on the floor up to the nozzles [28]. This portable and lighter structure cannot be too tall (normally a distance between 2 and 2.5 m from the nozzles to the ground). The water pumped through the nozzle is sprayed out as small drops with high speed (much higher than the terminal speed for those sizes in natural rain) because of the pressure added by the pump, which produces higher intensities and kinetic energy. Using the same type of nozzles, the drops produced are the same size, but a combination of different nozzles show a more realistic distribution. Depending on the height at which they are placed, different rainfall intensities and energies are obtained.

As can be seen in both types of rainfall simulators, there are issues that affect the characteristics defining the kinetic energy because the gravity-driven ones cannot reach the terminal speed, and the pressure-driven ones cannot produce drops equal to natural drops in mass. Thus, designing the best rainfall simulator for each purpose is essential, and it is critical to know the characteristics that a simulator must have to represent reality such as raindrop size distribution, raindrop impact velocity, rainfall intensity, kinetic energy, and random raindrop distribution [10,29].

In this paper, we present our design of a rainfall simulator, which has been specially designed to represent the kinetic energy of natural rainfall by controlling the production of drops with drop size distributions and speeds in the natural ranges.

## 2. Materials and Methods

For this project, the location of the simulator was an interior courtyard with a height of 10 m at the School of Engineering of the University of León, Spain. The maximum height, where the structures with the drippers were placed, allows the drops to reach terminal velocity [30–32]. This feature is one of the most difficult to achieve in laboratory simulators when simulating rainfall with adequate droplet sizes. The maximum droplet size that droplets can reach before breaking up and splitting in to two droplets is about 8 mm [31] and up to that size, they reach between 95% and 99% of their terminal velocity when they are produced from a height of around 10 m (Figure 1).

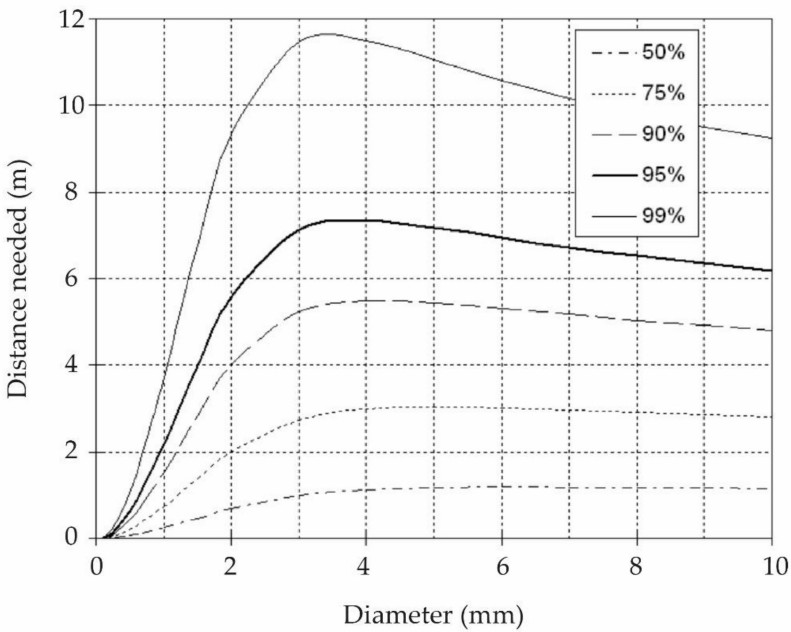

**Figure 1.** Graph showing the distance needed by raindrops to obtain the indicated percentage of their terminal velocity as a function of their diameter (taken from Van Boxel research [30]).

In addition, the vertical walls surrounding the location reduce the influence of the wind and prevent droplet dispersion.

The development of the simulator was implemented in several stages as improvements were made during the design to produce more realistic rainfall.

### 2.1. Simulator Structure Design

The simulator was designed as a 10 m high hanging structure to which water rises from a reservoir located at ground level. The first structure consists of four wooden slats forming a rectangle (Figure 2a). The space between the slats is covered by a metal mesh in which the drippers are placed. The water reaches the drippers through four independent paths that are connected to the hose through which the water rises from the pump. Each path has eleven spray nozzles in a random arrangement (Figure 2b). Two types of drippers have been used in this simulator, both with a flow rate range from 1 to 70 L/h. Some are drippers and others are nebulizers (Figure 2c) with a spray radius ranging from 0 to 0.5 m.

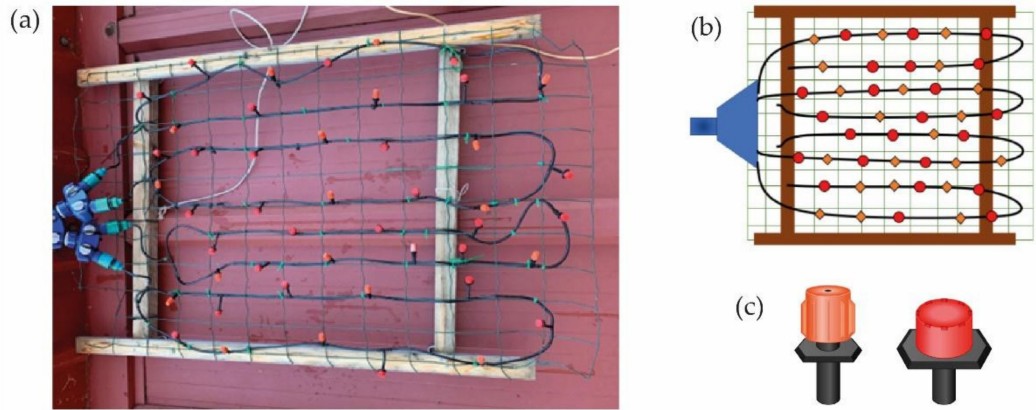

**Figure 2.** Figure showing the first wooden hanging structure: (**a**) photo of the wooden structure, (**b**) dripper placement scheme, and (**c**) types of drippers.

A second structure was also built with aluminum slats (Figure 3a). As it is a lighter material, the length of the structure can be longer. In this case, more resistant aluminum drippers were added, which provide a third type of drop (Figure 3b,c). The flow rate is the same as that of the drippers on the wooden structure, from 1 to 70 L/h.

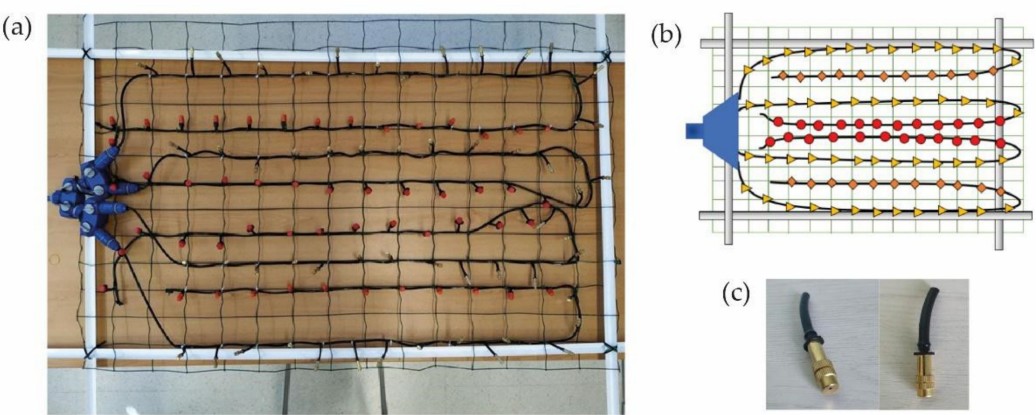

**Figure 3.** (**a**) Figure showing the second aluminum hanging structure, (**b**) dripper placement scheme, and (**c**) new type of dripper.

In each of the mini-sprinkler lines, different models of mini-nozzles were placed in order to increase the diversity of droplet sizes, taking into account the distribution of Gunn and Kinzer (1949) [32].

Both the wooden and aluminum platforms were mounted on rigid cables placed at the highest part of the interior courtyard in order to achieve sufficient height and adequate lateral wind protection.

As for the water supply, tap water was ruled out because the minerals it contains could alter the materials under study. To simulate rainwater, it is necessary to use distilled or deionized water, which was performed using a deionizer model Wasserlab DE7003.

A series of different pumps were used to move the water through the simulator depending on the test. The main pump, which is necessary in all cases, is the one that drives the water up to the hanging structure. It is an EH-125 HIDROBEX surface pump (Figure 4a). This pump drives the water from the 100 L tanks, which are connected to the main hose that reaches the hanging structure. A set of ball valves were strategically placed to increase the flow, thus modifying the flow rate, as well as automatic pressure gauges for flow control.

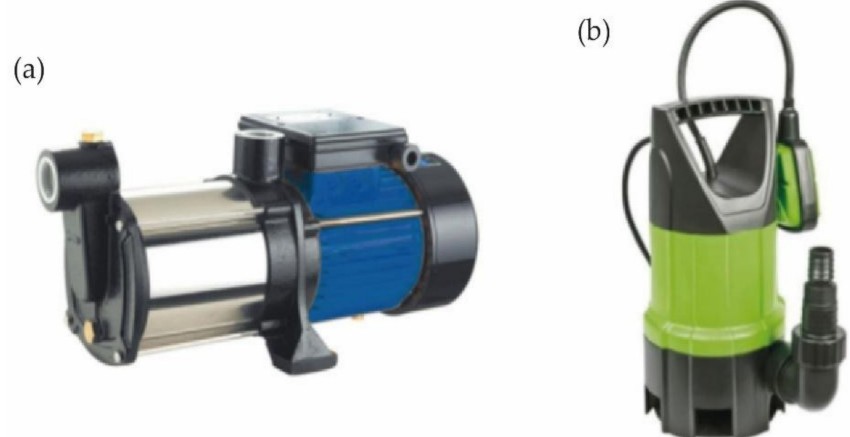

**Figure 4.** Pumps: (**a**) model EH-125 HIDROBEX and (**b**) model LISTA Hidrosub AS-216.

A hose attached to a water flow was used to fill the tanks connected to the main hose.

In one of the first phases of the development of the simulator, tanks were installed to be able to recover and recirculate the water. In order to recover the water that has dripped from the simulator onto the samples to be tested, a pool was installed. A second LISTA Hidrosub AS-216 bilge pump was used to recirculate the water (Figure 4b). This pump delivers water from the pool to the tank connected to the main hose.

In the later stages of the simulator development, the water was treated before it was introduced into the reservoir and could not be recirculated. In response, new tanks were added to treat the water. In this circuit, these new tanks are filled with deionized water from a hose connected to a water purification system. After treating the water, the bilge pump is introduced into the new tank and the water is transferred to the tank connected to the main hose to be pumped by the surface pump to the suspended structure (Figure 5).

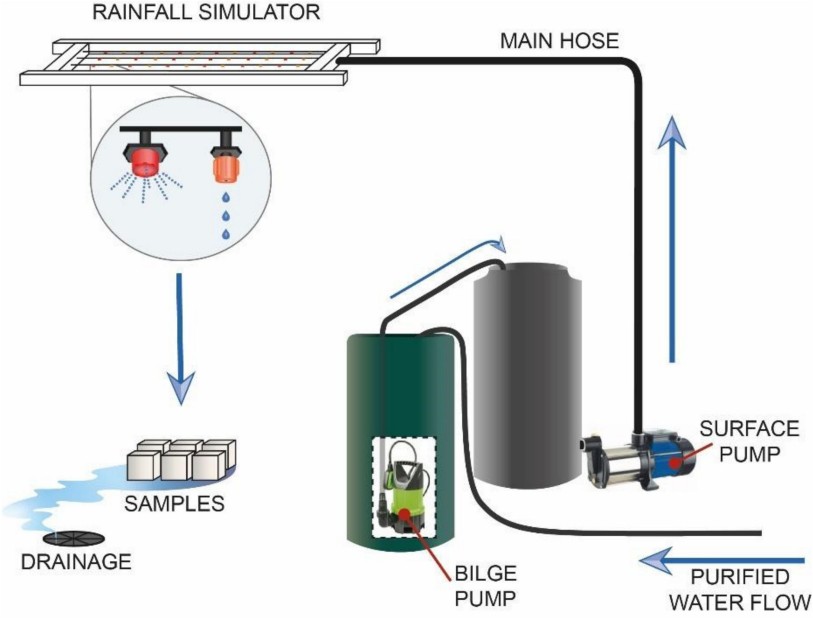

**Figure 5.** Simulator design.

After starting up the simulator for calibration (see next section), it was found that the pumps in charge of bringing water up the main hose lost power over time. Turning off the pump for a few seconds every minute solves this problem, so Arduino was used to program the pump operation. With this automated system, the surface pump runs for 55 s then stops for 5 s. This cycle can be programmed for the desired time.

### 2.2. Rainfall Simulator Calibration

In order to calculate the distribution of rainwater at each point of the selected sample-placement area, a first calibration of the rain simulator was carried out by placing 15 containers that collected rainwater in a given time. Containers had a capacity of 500 cm$^3$ and a diameter of 50 mm, and their distribution followed a pattern of concentric circles (Figure 6).

The largest circle has a diameter of 1.7 m for the aluminum structure and 1.2 m for the wooden structure. The circles were placed under their respective simulators. To perform the calibration, 2 min of rain was simulated, with a 5 s break at 55 s, and a total of 4 simulations were performed for each structure.

Calibrations are performed at the beginning of the data campaign. However, when temperatures drop, the pipes freeze and it is necessary to stop the tests, so these campaigns never last a year. This duration minimizes the possibility of finding significant differences over time due to simulator aging. In addition to these calibrations, during the whole campaign, in each simulation, 5 glasses are placed throughout the simulated area, which act as witnesses to check if the rainfall of each simulation has a similar intensity to the

previous simulation. For the moment, no significant changes have been observed within the same campaign that would require more frequent calibrations.

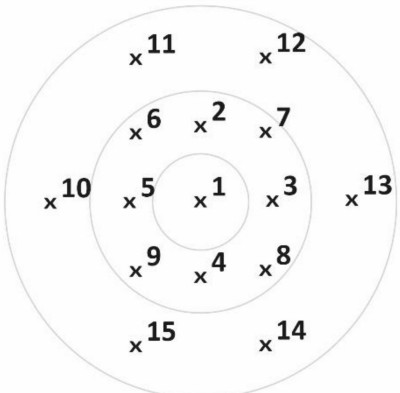

**Figure 6.** Placement of the fifteen containers in which the amount of water collected was measured.

In any case, whenever any part of the simulator has to be replaced, it is mandatory to perform the calibration again in order to redefine the intensity plane, distribute the subareas of similar intensities, and thus, be able to relocate the samples in those subareas to allow the rainfall received by each sample to be equivalent. To evaluate the uniformity of the calibration results, the mathematical equation for the Christiansen Uniformity Coefficient (CUC) was used. This method is widely used for calculating the uniformity of water application from sprinkler irrigation systems. The CUC may be expressed as:

$$CU = \left(1 - \frac{\sum_{i=1}^{N} |x_i - M|}{N * M}\right) * 100, \tag{1}$$

where $x_i$ is each result from $i = 1$ to $N$, $M$ is the results average, and $N$ in the number of results.

## 3. Results

Data obtained after measuring the volume of water contained in each of the fifteen containers after the four tests and the average for each container are shown in Tables 1 and 2.

**Table 1.** Calibration results for the simulator hanging from the wooden structure.

| Container No. | Test 1 (mL) | Test 2 (mL) | Test 3 (mL) | Test 4 (mL) | Mean (mL) |
|:---:|:---:|:---:|:---:|:---:|:---:|
| 1 | 41 | 44 | 51 | 53 | 47 |
| 2 | 46 | 40 | 57 | 58 | 50 |
| 3 | 52 | 39 | 55 | 55 | 50 |
| 4 | 36 | 22 | 38 | 32 | 32 |
| 5 | 14 | 11 | 13 | 16 | 14 |
| 6 | 8 | 11 | 10 | 10 | 10 |
| 7 | 37 | 43 | 41 | 38 | 40 |
| 8 | 39 | 29 | 34 | 39 | 35 |
| 9 | 9 | 10 | 14 | 11 | 11 |
| 10 | 11 | 10 | 13 | 6 | 10 |
| 11 | 8 | 10 | 9 | 8 | 9 |
| 12 | 18 | 10 | 18 | 16 | 16 |
| 13 | 18 | 12 | 19 | 18 | 17 |
| 14 | 35 | 20 | 29 | 26 | 28 |
| 15 | 5 | 4 | 10 | 9 | 7 |

**Table 2.** Calibration results for the simulator hanging from the aluminum structure.

| Container No. | Test 1 (mL) | Test 2 (mL) | Test 3 (mL) | Test 4 (mL) | Mean (mL) |
|:---:|:---:|:---:|:---:|:---:|:---:|
| 1 | 16 | 17 | 19 | 17 | 17 |
| 2 | 15 | 14 | 12 | 14 | 14 |
| 3 | 10 | 10 | 11 | 12 | 11 |
| 4 | 15 | 12 | 14 | 16 | 14 |
| 5 | 14 | 16 | 17 | 15 | 16 |
| 6 | 12 | 12 | 13 | 10 | 12 |
| 7 | 7 | 6 | 8 | 7 | 7 |
| 8 | 5 | 2 | 2 | 5 | 4 |
| 9 | 9 | 6 | 5 | 7 | 7 |
| 10 | 4 | 3 | 3 | 3 | 3 |
| 11 | 6 | 7 | 7 | 8 | 7 |
| 12 | 3 | 4 | 5 | 5 | 4 |
| 13 | 2 | 1 | 1 | 1 | 1 |
| 14 | 5 | 1 | 1 | 1 | 2 |
| 15 | 1 | 4 | 1 | 3 | 2 |

*Christiansen Uniformity Coefficient Results*

Since the aim is to simulate as much time as possible, the samples will be placed in the subareas where the greatest amount of water has been collected (Figure 7). In this way, only the calibration data in those subareas were taken into account for the calculation of the CUC.

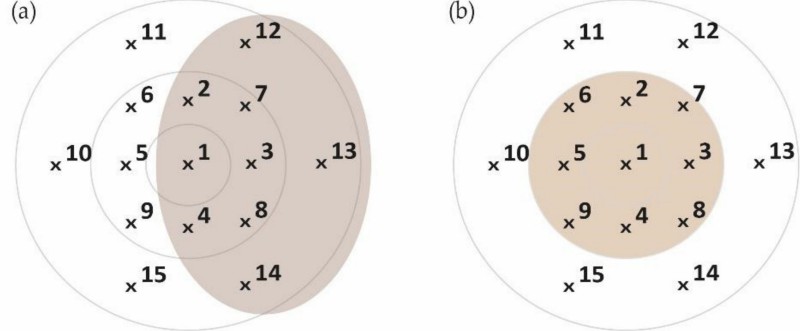

**Figure 7.** Subarea where the greatest amount of water was collected: (**a**) from wooden structure simulator and (**b**) from aluminum structure simulator.

In the case of the wooden structure, only the results of the positions 1 to 4, 7 and 8, and 12 to 14, represented by the shaded circle, were considered so $N = 9$ and $M = 35.0$. Thus, CUC is 69.68%. For the aluminum structure, positions considered were from 1 to 9, so $N = 9$ and $M = 11.2$. In this case, CUC is 66.83%. According to the American Society of Agricultural Engineers [33], the uniformity is poor for both cases (Table 3).

**Table 3.** Irrigation efficiency parameters classification according to the ASABE (1994).

| CLASSIFICATION | CUC (%) |
|:---:|:---:|
| Excellent | >90 |
| Good | 80–90 |
| Fair | 70–80 |
| Poor | 60–70 |
| Unacceptable | <60 |

A CUC value with percentage less than 80% implies that the spatial distribution is not good enough for evaluation of homogeneous areas of soil [34]. However, this is not a

problem when we use the rainfall simulator to evaluate small units, such as our samples of $5 \times 5$ cm$^2$, and we set samples all around the area because we may adapt the experiment to these conditions. Since we know that depending on the location the samples will receive different quantities of rainfall, by changing the position of the samples settled in the rainfall simulator to different positions, we may ensure that in the end, every sample has received a similar intensity and impact of rainfall. Indeed, this difference in intensity during the experiment can be seen as an advantage because it better represents the reality of the rainfall if the rainfall simulator is able to maintain the same intensity in each spot of the sampling area during every single event. So, the challenge is to show the temporal replicability, to ensure that during several simulated rainfall events, the characteristics at every point of the area remain stable.

In order to obtain the temporal distribution, the CUC of each container was calculated over time, from the first simulation to the fourth. The CUC values obtained for the position of each container are presented in Table 4.

**Table 4.** Christiansen Uniformity Coefficient results over time.

|        | Cont.* 1 | Cont. 2 | Cont. 3 | Cont. 4 | Cont. 5 | Cont. 6 | Cont. 7 | Cont. 8 | Cont. 9 | Cont. 10 | Cont. 11 | Cont. 12 | Cont. 13 | Cont. 14 | Cont. 15 |
|--------|------|------|------|------|------|------|------|------|------|------|------|------|------|------|------|
| CUC-W  | 89.9 | 85.6 | 88.8 | 84.4 | 88.9 | 91.0 | 94.3 | 89.4 | 86.4 | 80.0 | 91.4 | 87.5 | 85.8 | 83.6 | 64.3 |
| CUC-A  | 94.9 | 93.6 | 93.0 | 91.2 | 93.5 | 92.6 | 92.9 | 57.1 | 81.5 | 88.5 | 92.9 | 82.4 | 70.0 | 25.0 | 44.4 |

Note: * Cont.: Container. Position of the containers in Figure 7. Simulator hanging from the wooden structure (W) and from the aluminum structure (A).

Most of the values of the temporal distribution are good or excellent. Therefore, even if the amount of rainfall is not the same at each point, the average for an area will be the same every time. Thus, samples were separated into three groups and three zones with similar characteristics were delimited. Each group of samples spends the same time in each of the zones. In this way, a homogenous distribution of rainfall over each sample is achieved even if the spatial CUC result is poor.

As the simulator stages progressed, new calibrations were performed by placing the containers in positions that constituted a denser mesh of points. Thus, the extrapolation of the data between measurement points makes it possible to construct a surface that represents the distribution of rainfall intensity more accurately.

The last calibration was performed with 105 containers and the distribution of rainfall intensity can be seen in Figure 8.

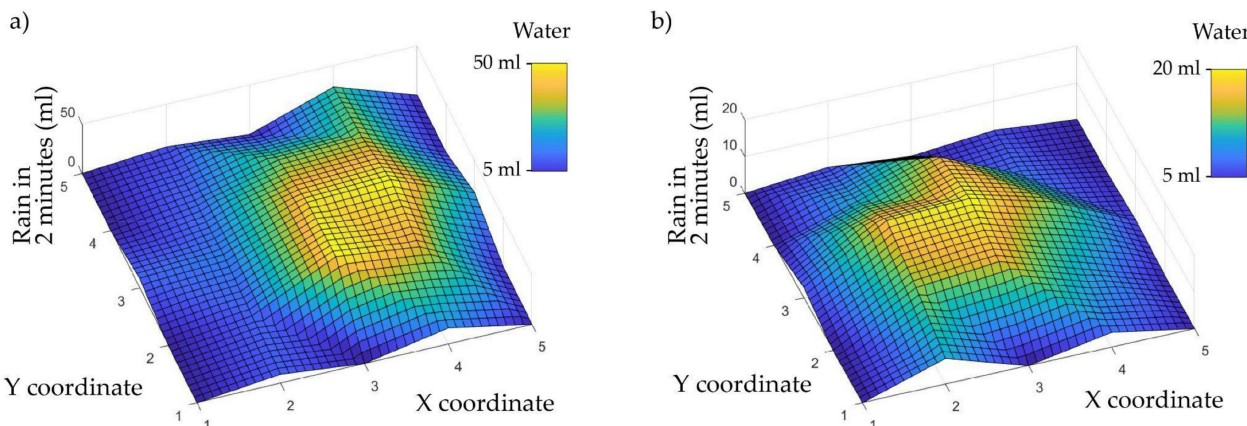

**Figure 8.** Rainfall intensity distribution under the (**a**) wooden and (**b**) aluminum structure simulator whose highest intensities mark the area to place the boxes with the samples.

The boxes with the rock samples of 125 cm$^3$ (5 cm $\times$ 5 cm $\times$ 5 cm) are placed within the area that receives the most characteristic precipitation of the represented climate and as

mentioned above, they are rotated to occupy three different positions within that area so that the mean precipitation is the same for all of them. The calibration results indicate that the maximum and minimum precipitation are 12.5 L/m$^2$ and 4 L/m$^2$, respectively, and the average precipitation in the selected area is 35 mL every 2 min. Therefore, precipitation is 8.91 L/m$^2$ per minute in the selected area or more than 500 L/m$^2$ per hour, which is classified as a heavy rain intensity (Table 5).

**Table 5.** Classification of the intensity of rainfall at any given time and place [35].

| CLASSIFICATION | L/m$^2$/h |
|---|---|
| Light | <3.5 |
| Moderate | 2.5–7.6 |
| Heavy | 7.6–50 |
| Torrential | >50 |

## 4. Discussion

The amount of water collected in the calibration containers has been used to compare the simulated precipitation on the samples with the mean annual precipitation calculated for León, Spain. The calculation was made with the total accumulated precipitation data recorded at the meteorological station of La Virgen del Camino (León) between 01 January 1990 and 01 January 2020 (AEMET) and with the data recovered by the disdrometer in the same campus where the rainfall simulator is ubicated [36,37].

The accumulated annual rainfall in León is approximately 500 mm so, knowing the rainfall data of the simulator, it is possible to calculate the time needed to simulate on the samples the rainfall of an entire year in the region of León. With a moderate rainfall intensity of 8.75 L/m$^2$, simulating one year's rainfall in the samples is possible. The approximate equivalence is that one hour of simulation is equivalent to one year of rainfall in León.

Knowing the equivalence in years of the simulator's operating time, it is possible to test different materials for durability, erosion, infiltration, etc., to evaluate the effects of rainfall over time. The simulator presented in this work is currently being used for a project in which the effectiveness of a coating applied on ornamental limestone rock is tested. Fifty years of rainfall in the province of León are being simulated and their effect on rock specimens with and without coating will be compared.

Considering that the nozzles are adjustable, the intensity of the simulated rainfall can be varied by adapting it to a specific study in a given region and/or season or for water-related durability tests in rocks, concrete, etc.

In this paper, a temporal homogeneity of rainfall intensity is pursued, in contrast to other works which use simulators to study the stability of the rainfall intensity focused in space [3,4,38,39]. The aim of this work is to obtain a temporal stability of precipitation in all the subareas of the experimental area. This method is similar to the one used by Zambon et al. [22]. Each subarea is characterized by its own intensity and drop size distribution, representing different natural events of the year in León. The individual samples are moved from one subarea to the next each hour of simulation (corresponding to one year of actual rainfall), so that all of them have experienced the whole range of rainfall intensities at the end of the simulation. The great achievement of the design of this simulator, thanks to its location, is to reach terminal velocity without sacrificing drop size, which makes it possible to reproduce more realistic rain conditions.

## 5. Conclusions

The success of this rainfall simulator resulted from the minimization of errors generated in conventional simulators. The main differential features are:



- Achieving a spectrum of mini sprinklers with droppers that produce droplet sizes similar to natural droplet sizes.
- Analyzing the distance required by a droplet to reach terminal velocity.
- Appropriate location that provides sufficient height to reach terminal velocity.
- Keeping the simulator sheltered from wind currents that may disperse water droplets.
- Versatility of the simulator to represent rainfall in different seasons, regions, or even in polluted atmospheres.
- Searching for homogeneity of results using the CUC.

There is a great interest in the scientific community to find a suitable water simulator that can represent conditions close to natural ones. We consider the simulator design presented in this work to be a good approximation for the construction of simulators that can be adjusted to represent the conditions of a certain region, as required.

**Author Contributions:** Conceptualization, M.F.-R. and I.R.; methodology, M.F.-R.; software, G.B. and P.C.; formal analysis, M.F.-R. and I.R.; investigation, G.B., A.O., R.M.-G. and P.C.; resources, R.M.-G. and A.O.; data curation, M.F.-R. and I.R.; writing—original draft preparation, I.R.; writing—review and editing, M.F.-R., I.R. and P.C.; supervision, M.F.-R.; project administration, M.F.-R.; funding acquisition, M.F.-R. All authors have read and agreed to the published version of the manuscript.

**Funding:** This research received funding from the Universidad de León (ULE-PoC 2018) and "Fundación General de la ULE y de la Empresa (FGULEM)" under project 2019/00149/001, from the "Consejería de Educación-Desafíos 2020" and the "Consejería de Turismo-PRESERVARTE project" of the Region of Castilla y León, and the granted MICINN project (PID2020-120439RA-I00).

**Informed Consent Statement:** "Not applicable" because it is a study that do not involve humans or animals.

**Data Availability Statement:** Not applicable.

**Acknowledgments:** To the rest of the Project team who help us every day and without whom it would not be possible to carry out the work: Covadonga Palencia, Fernando Jorge Fraile, Sagrario Fernández-Raga, Carlos Rodríguez, José Miguel González, Flavia Zelli, David González-Campelo, and Raúl García.

**Conflicts of Interest:** The authors declare no conflict of interest.

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
