# Peer review of "Optimization of a Laboratory Rainfall Simulator to Be Representative of Natural Rainfall"

_water, doi:10.3390/w14233831_

Round 1

Reviewer 1 Report

I read the work with great interest. In fact, it is certainly important, for the scientific community, an evaluation of a laboratory rainfall simulator to represent the natural rainfall. In my opinion the paper is readable, the methods are clearly stated and the results too. Perhaps the discussion section could be enlarged by comparing the results with those of other authors. For example, in Vinci et al., 2020, the microtopographic variations due to rainfall using a field rainfall simulator were assessed. In conclusion, in my opinion, the paper could be publishable under minor revision.

Author Response

Thank you very much for your comments. Here is a detailed answer to your great revision.

Reviewer 2 Report

The paper introduced a laboratory rainfall simulator with different nozzles of variable sizes of drops. It is an interesting topic, but some details need to be clarified in the current manuscript.

1           The description of gravity simulators and pressure simulators has been repeated many times which makes the Introduction a bit cumbersome. It is recommended to simplify the content and optimize the structure of the Introduction.

2     The references [1,3,4,6,8,9] had not been essentially quoted, and it is suggested to delete.

3     In this manuscript, the rainfall intensity distribution of this type of rainfall simulator in the test area was obtained through several calibrations. Then, according to the distribution , the samples to be tested can be placed at corresponding positions to simulate the required rainfall conditions. However, as time goes by, the simulator will inevitably have errors with the original distribution results due to natural aging or human factors, etc. Therefore, in order to ensure the accuracy of the simulator, it is necessary to calibrate a latest rainfall intensity distribution from the simulator before each test which could make the process of experiment more burdensom. Nevertheless, the rainfall simulator designed in this paper is indeed of practical significance.

Author Response

(The authors gave the same response as above.)

Reviewer 3 Report

The paper entitled, ‘Optimization of a laboratory rainfall simulator to be representative of natural rainfall’ presents important tools and results for an adequate field. The results match the contemporary issues in a good way. Only absence of the below comments.

·         Introduction section well describes the background what is known on the subject, and what elements are still subject to explore.

Lines 55-72 explains the advantages and disadvantages rainfall simulator classification based on working system (gravity or pressure-driven), and also Lines 94-111 explains the same thing which makes the introduction section too wordy, please check this.

·         Materials and Methods section will explain the Simulator structure design and Rainfall simulator calibration with appropriate illustrations (graphs & figures).

·         Result section will explain the outcomes of proposed methodology with appropriate illustrations.

·         Discussion section: please put your results in perspective with other reports in the literature, explain significance of results and how they contribute to the overall state of knowledge or how they advance knowledge.

·         Check accuracy of referencing style and please discuss more articles among international literature.

Author Response

(The authors gave the same response as above.)
